# Differences in Cell-Intrinsic Inflammatory Programs of Yolk Sac and Bone Marrow Macrophages

**DOI:** 10.3390/cells10123564

**Published:** 2021-12-17

**Authors:** Sara Elhag, Christopher Stremmel, Annette Zehrer, Josefine Plocke, Roman Hennel, Michaela Keuper, Clarissa Knabe, Julia Winterhalter, Vanessa Gölling, Lukas Tomas, Tobias Weinberger, Maximilian Fischer, Lulu Liu, Franziska Wagner, Michael Lorenz, Konstantin Stark, Hans Häcker, Marc Schmidt-Supprian, Uwe Völker, Martin Jastroch, Kirsten Lauber, Tobias Straub, Barbara Walzog, Elke Hammer, Christian Schulz

**Affiliations:** 1Medizinische Klinik und Poliklinik I, LMU Klinikum, Ludwig-Maximilians-Universität, 81377 Munich, Germany; sara-r.m.elhag@hotmail.com (S.E.); clarissa.knabe@outlook.de (C.K.); julia.winterhalter@med.uni-muenchen.de (J.W.); lukas.tomas@med.uni-muenchen.de (L.T.); tobias.weinberger@med.uni-muenchen.de (T.W.); maximilian.fischer@med.uni-muenchen.de (M.F.); lulu.liu@med.uni-muenchen.de (L.L.); franziska.wagner@med.uni-muenchen.de (F.W.); michael.lorenz@med.uni-muenchen.de (M.L.); konstantin.stark@med.uni-muenchen.de (K.S.); 2DZHK (German Centre for Cardiovascular Research), Partner Site Munich Heart Alliance, 80336 Munich, Germany; 3Biomedical Center, Institute of Cardiovascular Physiology and Pathophysiology, Ludwig-Maximilians-University Munich, Planegg-Martinsried, 82152 Munich, Germany; annette.zehrer@web.de (A.Z.); walzog@lrz.uni-muenchen.de (B.W.); 4Walter Brendel Center of Experimental Medicine, LMU Klinikum, Ludwig-Maximilians-Universität, 81377 Munich, Germany; 5Department of Functional Genomics, Interfaculty Institute for Genetics and Functional Genomics, University Medicine Greifswald, 17475 Greifswald, Germany; josefine.plocke@googlemail.com (J.P.); voelker@uni-greifswald.de (U.V.); hammer@uni-greifswald.de (E.H.); 6Department of Radiation Oncology, LMU Klinikum, Ludwig-Maximilians-Universität, 81377 Munich, Germany; roman.hennel@med.uni-muenchen.de (R.H.); kirsten.lauber@med.uni-muenchen.de (K.L.); 7Department of Molecular Biosciences, The Wenner-Gren Institute, Stockholm University, 106 91 Stockholm, Sweden; michaela.keuper@su.se (M.K.); martin.jastroch@helmholtz-muenchen.de (M.J.); 8Institute of Experimental Hematology, School of Medicine, Technical University of Munich, 81675 Munich, Germany; vanessa.goelling@tum.de (V.G.); marc.supprian@tum.de (M.S.-S.); 9Center for Translational Cancer Research (TranslaTUM), School of Medicine, Technical University of Munich, 81675 Munich, Germany; 10Division of Microbiology and Immunology, Department of Pathology, University of Utah, Salt Lake City, UT 84112, USA; hans.haecker@path.utah.edu; 11DZHK (German Centre for Cardiovascular Research), Partner Site Greifswald, 17475 Greifswald, Germany; 12German Cancer Consortium (DKTK), Partner Site Munich, 80336 Munich, Germany; 13Core Facility Bioinformatics, Biomedical Center, Ludwig-Maximilians-University Munich, Planegg-Martinsried, 82152 Munich, Germany; tstraub@bmc.med.lmu.de

**Keywords:** macrophages, yolk sac, inflammasome

## Abstract

Background: Tissue-resident macrophages have mixed developmental origins. They derive in variable extent from yolk sac (YS) hematopoiesis during embryonic development. Bone marrow (BM) hematopoietic progenitors give rise to tissue macrophages in postnatal life, and their contribution increases upon organ injury. Since the phenotype and functions of macrophages are modulated by the tissue of residence, the impact of their origin and developmental paths has remained incompletely understood. Methods: In order to decipher cell-intrinsic macrophage programs, we immortalized hematopoietic progenitors from YS and BM using conditional HoxB8, and carried out an in-depth functional and molecular analysis of differentiated macrophages. Results: While YS and BM macrophages demonstrate close similarities in terms of cellular growth, differentiation, cell death susceptibility and phagocytic properties, they display differences in cell metabolism, expression of inflammatory markers and inflammasome activation. Reduced abundance of PYCARD (ASC) and CASPASE-1 proteins in YS macrophages abrogated interleukin-1β production in response to canonical and non-canonical inflammasome activation. Conclusions: Macrophage ontogeny is associated with distinct cellular programs and immune response. Our findings contribute to the understanding of the regulation and programming of macrophage functions.

## 1. Introduction

Macrophages are key effectors of innate immunity. They contribute to organ functions, the maintenance of tissue homeostasis and defense against pathogenic invasion [1]. In order to fulfill organ-specific functions, resident macrophages adopt distinct properties, e.g., clearance of surfactant by alveolar macrophages or breakdown of bone tissue by osteoclasts [2]. These functions are largely imprinted by the tissue environment. Consequently, adoptive transfer of mature macrophages between organ compartments results in their rapid molecular and phenotypic adaptation to the new microenvironment [3,4]. Despite the strong impact of the local milieu on macrophage identity, significant heterogeneity can be observed between individual macrophages inside the same tissue [5]. This heterogeneity might be explained in part by different developmental origins of macrophage populations. Except for microglia in the central nervous system, which are solely derived from yolk sac (YS) hematopoiesis in early life, macrophages in mouse tissues are derived in part from embryonic hematopoiesis as well as from definitive hematopoiesis of the bone marrow (BM) [4,6,7,8,9]. The impact of ontogeny on the cellular identity of macrophages remains unclear.

In this study, we isolated hematopoietic progenitors of YS- and BM-derived macrophages and conditionally immortalized them through expression of the transcription factor Hoxb8. We then differentiated Hoxb8 progenitors into mature macrophages under identical cellular conditions in vitro and characterized their cellular identity by deep immunophenotyping and functional assays. While the phenotypic appearance and basic cellular functions of YS- and BM-derived macrophages were similar, we identified differences in their cell-intrinsic programs, such as their metabolic response to inflammatory cues. Further, BM-derived macrophages exhibited strong activation of the NLR family pyrin domain containing 3 (NLRP3) inflammasome pathway when exposed to crystal particles, leading to robust interleukin-1 (IL-1) β production, whereas both canonical and non-canonical inflammasome responses were blunted in YS-derived macrophages. Our study contributes to defining the cellular identity of YS- and BM-derived macrophages in the absence of tissue-specific environments.

## 2. Materials and Methods

### 2.1. Mice

C57BL/6J CD45.1 (Ptprca) and CD45.2 (Ptprcb) congenic mice were purchased from Jackson Laboratories. Casp1/Casp11 double knockout mice have been described previously [10].

### 2.2. Generation of Yolk Sac and Bone Marrow Derived ER-Hoxb8 Progenitors

HEK-293T (CRL-3216™ ATCC, Manassas, VA, USA) cells were transfected with both ecotropic packaging vector pCL-Eco and retroviral backbone pMSCVneo-ER-Hoxb8 using Lipofectamin (Invitrogen 11668-019, Waltham, MA, USA). Virus-containing supernatant was harvested 48 h post transfection, filtered using 0.8 µm cellulose acetate filters (Sartorius 16592, Gottingen, Germany), and either used immediately or aliquoted and stored at −80 °C. The virus was titrated using NIH-3T3 cells, and CFU per milliliter was determined.

BM derived hematopoietic progenitors were isolated from 8–12-week-old mice, and Hoxb8 transfection was performed as described previously [11]. For the generation of the YS progenitor cell lines, timed matings were scheduled and YS membranes from mouse embryos were isolated at E9.5, as described below. Embryos were removed from the uterus and washed in 4 °C phosphate buffered saline. The YS membrane was harvested and digested in PBS containing 1 mg/mL collagenase D (Roche 11088866001, Basel, Switzerland), 100 U/mL desoxyribonuclease I (Sigma-Aldrich D7291, St. Louis, MO, USA) and 1% FCS (BIO&SELL S0615, Feucht-Nürnberg, Germany), at 37 °C for 15 min. Tissues were mechanically dissociated and passed through a 100 μm cell strainer. Immediately after tissue digestion, progenitor cells were transduced with the ER-Hoxb8 encoding retrovirus by spinoculation with Polybrene (TR-1003, Sigma-Aldrich, Taufkirchen, Germany) in progenitor outgrowth medium (proliferation medium) consisting of RPMI 1640 (Sigma R 8758) supplemented with non-heat inactivated 10% FCS (BIO&SELL S0615), 1% penicillin–streptomycin (Sigma P 4333), 1μM β-estradiol (Sigma E2257), 30 μM 2-Mercapto-Ethanol (Sigma M3148) and 6% SCF-containing supernatant (80 min at 1500 g, room temperature). After 4 weeks of culture in proliferation medium the immortalized Hoxb8-SCF cell line was established.

### 2.3. Generation of SCF Supernatant

In a T175 cell culture flask, 10^6^ SCF-producing cells (CHO-MGF) were cultured in 30 mL RPMI 1640 (Sigma R8758) supplemented with 10% FCS (BIO-SELL S0615) and 1% penicillin–streptomycin (Sigma P 4333). The medium was collected 1–2 days after confluency and passed through a 0.4 µm sterile filter prior to freezing at −20 °C for future use.

### 2.4. Generation of FLT3 Supernatant

In a T175 cell culture flask, 10^6^ FLT3-producing B16-FLT3L (RRID:CVCL_IJ12) cells were cultured in 50 mL RPMI 1640 (Sigma R8758) supplemented with 10% FBS (F0804, Sigma). The medium was collected 1 day after confluency was reached and passed through a 0.4 µm sterile filter prior to freezing at −20 °C for future use. FLT3 concentration was determined using a Mouse/Rat Flt-3 Ligand Quantikine ELISA Kit (R & D Systems MFK00, Minneapolis, MN, USA).

### 2.5. Differentiation of ER-Hoxb8 Progenitors

BM ER-Hoxb8 progenitor cells and 5 × 10^5^ YS were seeded per 10 cm tissue culture dish in 5 ml of differentiation medium (RPMI 1640 with 10% FCS (BIO-SELL S0615), 1% penicillin–streptomycin (Sigma P 4333), 30 μM 2-Mercapto-Ethanol (Sigma M3148) and supernatant containing 6% SCF or FLT3, respectively, and 10 ng/mL M-CSF (Immuno Tools 12343115, Friesoythe, Germany)) at 37 °C/5% CO_2_. Cell culture medium was changed every second day till the intended experiments were performed. Bright field images during the process of differentiation were collected with a Zeiss (Jena, Germany) Axiovert 200, Axiocam HRC microscope. Cells were counted every day at the same time for 6 days, and doubling time was calculated.

### 2.6. Detachment of Differentiated Adherent Cells

Differentiated macrophages were washed twice with cold (4 °C) Dulbecco’s Phosphate Buffered Saline (Sigma-Aldrich P5493) and incubated at 37 °C in 2 mL Accutase (Sigma A6964) for 5–10 min with slight agitation until full cell detachment was achieved. If detachment was incomplete, cells were gently detached using a cell scrapper. Subsequently, detached cells were washed with cell culture medium and kept on ice for further processing.

### 2.7. Flow Cytometry

Cells were centrifuged at 400 *g* for 5 min, resuspended in 4 °C PBS (Sigma-Aldrich P5493), plated in multi-well round-bottom plates and immunolabeled for FACS analysis. Desired antibody mixtures were added to 1% BSA/PBS and incubated for 20 min at 4 °C. Flow cytometry evaluation was performed using a BD Biosciences LSR Fortessa flow cytometer, and data were analyzed using FlowJo 10.

### 2.8. May–Grünwald–Giemsa Staining

Cells were seeded and differentiated in a 1-well glass slide (Nunc^®^ Lab-Tek^®^ II Chamber Slide™ system, Thermo Fisher Scientific, Waltham, MA, USA). On day 5, slides were placed in May–Grünwald–Giemsa (Sigma-Aldrich 63590) staining solution for 5 min, then washed in Phosphate Buffer pH 7.2, for 90 s, followed by Giemsa staining solution (Sigma-Aldrich GS500) (diluted 1:20 with deionized water) for 15–20 min. Slides were rinsed in deionized water, air-dried and analyzed with a Zeiss Imager M2 microscope.

### 2.9. Immunofluorescence Staining

For immunofluorescence, cells were plated and differentiated on 8-well chamber slides (Nunc^®^ Lab-Tek^®^ Chamber Slide™ system). On day 5, the differentiation medium was decanted and slides were washed, fixed with 4% PFA (Carl Roth 0335.3, Karlsruhe, Germany) for 10 min and washed again. Blocking was performed with 10% donkey serum for 30 min. Slides were stained with primary antibodies, namely, anti-KI67 (ab15580), anti-CX3CR1 (702321) and anti-F4/80 (ab6640) followed by secondary antibodies generated in their respective hosts for 1 h each. Nuclei were counterstained with Hoechst (Invitrogen H3569). Slides were mounted using Ibidi (Gräfelfing, Germany) mounting medium for fluorescence microscopy (Ibidi 50001) and visualized using a ZEISS Axio Imager M2 microscope. Fluorescence intensities were analyzed using ImageJ Version 1.53k.

### 2.10. Phagocytosis Assay

YS and BM ER-Hoxb8 progenitor cells were plated and differentiated on 8-well chamber slides. On day 5, the differentiation medium was decanted, and slides were washed with sterile PBS. Subsequently, 100 µL of pHrodo™ Green Zymosan Bioparticles™ Conjugate for Phagocytosis (Thermo Fisher Scientific) was added per well and incubated for 60 min at 37 °C. Slides were washed with PBS, fixed using 4% PFA for 10 min and washed again before staining for F-actin with Invitrogen™ Texas Red™-X Phalloidin (Invitrogen T7471). Nuclei were counterstained with Hoechst (Invitrogen H3569). Slides were visualized using a ZEISS LSM 880 microscope and a ZEISS Axio Imager M2. To compare phagocytosis efficiency, an average of 6 representative images from 3 independent experiments was evaluated using ImageJ. The total number of macrophages was determined by DAPI expression and phagocytic macrophages were identified by GFP expression, and the ratio of GFP+ to DAPI+ macrophages was calculated.

### 2.11. Intraperitoneal Injection and Differentiation of Hoxb8 Progenitor Cells In Vivo

For analysis of in vivo survival and differentiation, intraperitoneal injection of 107 YS Hoxb8 cells was performed. Cells were pre-differentiated with M-CSF (Immuno Tools 12343115) for three days with M-CSF (as described above) prior to injection. Cell suspension was applied in 400 µL sterile PBS with a 27 G needle after prewarming by hand. After one week, the mice were euthanized by cervical dislocation and cells were recovered by peritoneal lavage using 10 mL ice cold PBS with 3% FCS and prepared for subsequent FACS analysis.

### 2.12. Apoptosis and Necrosis Assays

Hoxb8 progenitors were differentiated for 5 days, cells were detached by cell scraping and seeded at a density of 20,000 cells in 3 mL of differentiation medium in each well of 12-well plates. Macrophages were treated with different doses of SuperFasLigand (ALX-522-020-C005, Enzo Life Sciences, Farmingdale, NY, USA) or UV irradiation on a UV Stratalinker 1800 (Stratagene, San Diego, CA, USA), respectively. At the indicated time points, cell culture supernatants were collected and macrophages were detached by Accutase (Sigma A6964) treatment. The corresponding cell culture supernatants and detached macrophages were combined and washed with annexin V binding buffer (BD Biosciences 556454). Afterwards, macrophages were stained with annexin V-FITC (BD Biosciences 556547) and 2 µg/mL propidium iodide (Sigma-Aldrich CAS 25535-16-4) in 100 µL annexin V binding buffer for 15 min on ice. Subsequently, stained macrophages were washed with annexin V binding buffer and subjected to flow cytometry on an LSR II cytometer (BD Biosciences). Annexin V-FITC-positive, propidium iodide-negative macrophages were considered as apoptotic, while annexin V-FITC and propidium iodide-double-positive macrophages were considered as necrotic.

### 2.13. Extracellular flux XF96 Seahorse Measurements

Differentiated Hoxb8 macrophages were detached by cell scraping and seeded at a density of approximately 105 cells per well (in 200 µL differentiation medium) in a Seahorse 96-well plate. On day 5, medium was removed and cells were incubated with and without the following stimuli: 100 ng/mL LPS (Sigma L8274) or 100 ng/mL IL4 (Immuno Tools 12340043). After 4 h incubation time, medium was replaced with 180 µL/well XF assay medium (Agilent 102365-100) supplemented with 11 mM glucose (Sigma G6152) (pH adjusted to 7.5) and incubated for 60 min in an air incubator. The XF96 plate (Seahorse Bioscience, Agilent Technologies, Santa Clara, CA, USA) was then transferred to a temperature-controlled (37 °C) Seahorse extracellular flux analyzer (Agilent Technologies) and subjected to an equilibration period. One assay cycle comprised a 1 min mix, 2 min wait and 3 min measure period. Oxygen consumption rates (OCR) and extracellular acidification rates (ECAR) were analyzed and dissected into different functional modules, as described in detail previously [12]. In short: after 4 basal assay cycles, oligomycin (1 µg/mL) (Sigma O4876) was injected to inhibit the ATP synthase to determine OCR related to ATP synthesis for 3 cycles. Then, 2,4-Dinitrophenol (DNP; 100 µM) (Sigma 34334) was injected to stimulate maximal respiration by protonophoric action, indicating maximal substrate oxidation rates (3 cycles). Next, pyruvate (5 mM) (Sigma P8574) was injected to remove rate-limitation by glycolysis (3 cycles). Rotenone (R, 4 µM) (Sigma R8875) plus antimycin A (AA, 2 µM) (Sigma A8674) was added followed by 4 assay cycles to determine the non-mitochondrial OCR. The mean of 4 OCR measurements after addition of R/AA was subtracted from all other rates. To determine extracellular acidification rates (ECARs) deriving from glycolysis, the last injection also contained 2-deoxy-glucose (2 DG, 100 mM). The mean of 4 ECAR measurements after 2 DG injection was subtracted from all ECAR values to obtain ECAR due to glycolysis. Mitochondrial efficiency was determined by calculating coupling efficiency (CE) (the fraction of basal mitochondrial respiration that is linked to ATP synthesis) and by calculating the cellular respiratory control ratio cRCR (maximal respiration divided by proton leak). After the measurement, cells were lysed and total dsDNA quantities per well were determined using a Quant-iT PicoGreen dsDNA Assay Kit (Thermo Fisher Scientific). All rates were normalized to 130 ng dsDNA (=mean DNA content/well of all measurements).

### 2.14. Multiplex Cytokine Immunoassay

Hoxb8 macrophages were differentiated in 6-well plates for 5 days with 5 × 10^4^ cell per well. Medium was changed on days 3 and 4. On day 5, medium was replaced with either 2 mL of differentiation medium only, or differentiation medium with stimuli, as follows, with 100 ng/mL LPS (Sigma L8274) or 100 ng/mL IL4 (Immuno Tools 12340043). Three wells per condition were used and supernatants were pooled. To determine cytokine concentrations in cell supernatants, the Bio-Plex Pro Mouse Cytokine Assay (BIO-RAD LABORATORIES 171G6005M, 171G6006M and 171G6009M) was used on a Luminex 200 system according to the manufacturer’s protocol (Bio-Rad Laboratories, Munich, Germany).

### 2.15. Inflammasome Activation

Differentiated Hoxb8 cells were washed and incubated in 5 mL of differentiation medium supplemented with 200 ng/mL LPS (Sigma L8274) for 3 h. Then, 250 µg/mL MSU, 375 ug/mL cholesterol crystals or 2 µg/mL *E. coli* OMVs were added and incubated for indicated time periods. Supernatant was collected and IL1β was measured using Mouse IL-1 beta/IL-1F2 DuoSet ELISA (R & D Systems DY401-05) according to the manufacturer’s instruction. Tecan GENios was used for OD evaluation; concentrations were calculated with standard curve (detection limit, 15.6 pg/mL).

### 2.16. Western Blotting

Cell were harvested in 500 µL of radioimmunoprecipitation assay buffer (RIPA buffer) (Sigma-Aldrich R0278) supplemented with Halt™ Protease Inhibitor Cocktail (100×) (Thermo Fisher Scientific PIER87786) and collected in 1.5 mL Eppendorf tubes. After incubation for 30 min on ice, lysates were centrifuged for 20 min at 14,000 *g*. Supernatant was collected and protein amount was estimated using the Pierce™ BCA Protein Assay Kit (Thermo Fisher Scientific 23225) according to the manufacturer’s instructions. Then, 10 µg amount of total protein was loaded per lane on a 10%/4–12% NuPAGE Bis-Tris gel (Thermo Fisher Scientific NP0301BOX and NP0322PK2). After electrophoresis, proteins were transferred to a nitrocellulose membrane. After blocking with 5% non-fat milk for 1 h at room temperature, the membrane was washed and incubated with desired primary antibodies at 4 °C overnight. After washing, secondary antibodies were incubated for 1 h at room temperature. For final evaluation, the membrane was washed again and incubated with ECL western blot substrate (Thermo Fisher Scientific 32209), before transfer to an Amersham ImageQuant 800 Western blot imaging system. Membranes were incubated in Restore™ Plus Western Blot Stripping Buffer (Thermo Fisher Scientific 10016433) for 5 min, followed by a washing step, and then blocked with 5% non-fat milk for 1 h at room temperature, before incubation with indicated primary antibodies.

### 2.17. Lysate Collection for Transcriptome and Proteome Analyses

Hoxb8 progenitor cells were differentiated for 5 days. Tissue culture plates were then washed twice with PBS. For stimulation experiments, cells were incubated for 5 h in 5 mL of differentiation medium supplemented with 100 ng/mL LPS (Sigma L8274) or 100 ng/mL IL4 (Immuno Tools 12340043) as indicated. After stimulation, cells were washed, pelleted and resuspended in 500 µL TRIzol (Sigma T9424) (for transcriptome analyses) or 500 µL of urea/thiourea buffer (for protein analyses) in low protein binding 1.5 mL Eppendorf tubes.

### 2.18. Protein Profiling by Mass Spectrometry

Sample preparation and mass spectrometric analyses by liquid chromatography coupled tandem mass spectrometry (LC–MS/MS) using a LTQ-Orbitrap Velos instrument were carried out as described previously [13]. For all conditions, four bioreplicates from independent experiments were analyzed. In short: protein was extracted by multiple freeze-thaw cycles and collected by 1 h centrifugation (room temperature, 19.000 *g*) after nucleic acid fragmentation with a sonication probe. For each sample, 3 µg protein was reduced (DTT) and alkylated (iodoacetamid) before digestion with trypsin at a protein to enzyme ratio 25:1 (37 °C, 16 h). Peptide mixtures were desalted on C18 material (µZipTip, Merck Millipore, Burlington, MA, USA). Peptides were separated by LC (nano-Acquity UPLC system, Waters, Milford, MA, USA) before data-dependent acquisition of MS data. MS spectra were acquired in the Orbitrap whereas fragment spectra (MS2) of the 20 most abundant ions were recorded in a linear ion trap (LTQ).

Mass spectrometric raw data was searched against a mouse SwissProt database (16 September 2016). Identification and comparative quantification of proteins in YS and BM macrophages was carried out in Progenesis QI (Nonlinear Dynamics, Micromass UK Limited, Newcastle upon Tyne, UK) via a Mascot search algorithm (v2.3, Matrix Science, London, UK). For protein quantitation, Hi3 non-conflicting peptides (score > 20) were considered. For data analysis of cells stimulated with LPS in comparison with untreated cells, peptide/protein identification at an FDR of 1% was carried out with the Andromeda algorithm implemented in MaxQuant v1.5.3.8. Normalization for differences in peptide loading was carried out using the MaxQuant LFQ algorithm for label-free quantification [14]. Resulting protein intensities (Label Free Quantification values (LFQ)) were exported and statistically analyzed.

### 2.19. Gene Expression Analyses

RNeasy Micro Kit (Qiagen 74004, Hilden, Germany) was used for RNA isolation. RNA was then transcribed to cDNA using a High-Capacity cDNA Reverse Transcription Kit (Applied Biosystems 10400745, Waltham, MA, USA) with RNase Inhibitor (Thermo Fisher Scientific EO0381). Quantification of cDNA was performed with desired primers (see Key Resource Table) by real-time polymerase chain reaction with SsoAdvanced Universal SYBR Green Supermix (BIO-RAD 64296124) using a StepOnePlus Real-Time PCR System.

### 2.20. RNA Sequencing and Data Analysis

RNA quantification and purity: An aliquot of each total RNA sample was used to determine RNA concentration and purity on the NanoDrop ND-1000 spectral photometer (Thermo Fisher Scientific). Furthermore, all samples were analyzed on the 2100 Bioanalyzer (Agilent Technologies) using RNA Nano/HS LabChip Kits (Agilent Technologies) to determine RNA integrity.

Library preparation: Library preparation was performed with the Illumina (San Diego, CA, USA) TruSeq^®^ Stranded mRNA technology, according to the manufacturer’s protocol. The protocol started with an RNA fragmentation step using divalent cations. Then, reverse transcription was carried out to generate first strand cDNA. In the second strand cDNA synthesis dUTP is incorporated instead of dTTP to avoid amplification by DNA polymerase in the subsequent PCR and thus guarantee strandedness. The 3′-ends were then adenylated and sequencing adapters were ligated. These comprised sequencing primer- and flow cell-binding sites, as well as indices for multiplexed sequencing of pooled libraries. Adapter-ligated fragments were amplified during a limited-cycle PCR reaction. After the limited cycle PCR, at the end of the library preparation, all samples were quality controlled. DNA 1000/HS LabChip kits were used with the 2100 Bioanalyzer (Agilent Technologies). Furthermore, all libraries were quantified using the highly sensitive fluorescent dye-based Qubit ds DNA HS Assay Kit (Thermo Fisher Scientific). In brief, 1 μL of each sample was used to determine ds DNA concentration (ng/μL) in comparison to a given standard provided with the kit. The DNA concentration was determined by creating a linear trend line and applying the mathematical equation of the linear regression. All single libraries were pooled into a sequencing library with an equal DNA amount per sample. After quantification, the final sequencing library was diluted to 2.25 nM, followed by denaturation with NaOH to ensure the presence of single-stranded DNA fragments for cluster generation.

Cluster generation and sequencing: Cartridge loading was conducted following the manufacture’s recommendations for NovaSeq 6000, according to the standard workflow using a SP flowcell. Template amplification and clustering was performed onboard the NovaSeq 6000, applying exclusion amplification (ExAmp) chemistry. For cluster generation and subsequent sequencing of all samples, one single-read 75 cycle run was performed, on an SP flowcell. Cluster generation and sequencing were operated under the control of the NovaSeq Control Software (NVCS) v1.6.0. After cluster generation, sequencing primers hybridize to the adapter sequences at the end of the fragments and sequencing is carried out. Sequencing was performed with reads of a length of 75 bp (single-read).

Data analysis: If not indicated otherwise, default parameters were used for data processing. RNA sequencing reads were aligned to the mouse genome (ENSEMBL release GRCm38.94) and counted per gene using STAR (version 2.6.1d). Transcripts per million (TPM) estimates were obtained using RSEM (version 1.3.0). Differential expression analysis was performed in R/bioconductor with DESeq (1.26.0).

### 2.21. Statistical Analysis

All experiments were performed with at least two independent clones per cell line. Extracellular flux assays were evaluated by two-way ANOVA, followed by a Sidak post hoc test. Inflammasome experiments were evaluated by one-way ANOVA with a Bonferroni post hoc test. Comparisons between other groups were calculated using unpaired, two-tailed *t*-tests, (***) *p* < 0.001, (**) *p* < 0.01, (*) *p* < 0.05. Errors bar indicate mean +/− standard deviation or median with interquartile range (IQR), as indicated. All graphs and calculations were generated with GraphPad Prism 7 software.

## 3. Results

### 3.1. Immortalization of YS and BM Hematopoietic Progenitors Using Conditional Hoxb8

In order to compare YS and BM derived macrophages independently of their local environment, we isolated hematopoietic progenitors from embryonic day (E) 9.5 YS and BM of 8–12-week-old mice. Cells were transduced with an estrogen-regulated Hoxb8 (ER-Hoxb8), allowing their maintenance and expansion at the progenitor level [11,15]. Upon estrogen removal, both cell lines differentiated to mature macrophages in the presence of macrophage colony-stimulating factor (M-CSF) (Figure 1a).

### 3.2. Hoxb8 YS Progenitors Derive from KIT+ Hematopoietic Cells

Previous studies reported successful conservation of the hematopoietic progenitor stage with the Hoxb8 system, but Hoxb8 cell lines of YS macrophage progenitors have not been reported to date. Thus, we first examined primary YS cells in more detail. We analyzed single-cell suspensions, which were directly isolated from E9.5 YS, by flow cytometry for surface expression of progenitors as well as macrophage differentiation markers. In line with previous studies, we identified KIT+ erythromyeloid progenitors in the YS and confirmed Kit expression by RT-PCR (Figure 1b,c) [16,17,18]. We then confirmed Kit expression also in immortalized Hoxb8 cells from YS and BM (Figure 1d). In summary, the Hoxb8 system successfully conserves the progenitor stage of cells isolated from the YS as well as BM.

### 3.3. Cell Growth, Expansion and Phagocytic Capacity of YS and BM Macrophages Are Similar under Defined Conditions

Next, we compared cell growth and differentiation between YS and BM Hoxb8 cells. Proliferation was similar in both populations with doubling times of approximately 16 h (Figure 1e). Upon estrogen withdrawal, YS- and BM-derived Hoxb8 progenitors differentiated in a similar fashion, reaching mature macrophage morphology around day 5 (Figure 1f). This typical cell morphology was also apparent with May–Grünwald–Giemsa stainings and characteristic for the mature phenotype in both cell types (Figure 1g). Markers of early progenitor states, namely, Kit and Runx1, were downregulated in BM and YS Hoxb8 macrophages during maturation. In contrast, markers of macrophage differentiation, such as Lyz2, Csf1r and Cx3cr1, were upregulated in both populations (Figure 1d). Analogue expression patterns for KI67, as well as leukocyte (CD45), myeloid (CD11b) and macrophage (F4/80, CSF1R) markers, were confirmed at the protein level by immunofluorescence and flow cytometry (Figure 1h,i and Figure 2b).

Differentiated YS and BM Hoxb8 macrophages exhibited efficient phagocytotic capacity when exposed to Zymosan bioparticles. After lysosomal fusion, engulfed Zymosan bioparticles turned green (pHrodo) indicating successful phagocytosis and phagolysosomal acidification. In immunofluorescence analyses, both cell types showed comparable amounts of engulfed particles after 60 min incubation (Figure 1j,k).

### 3.4. Macrophage Differentiation

After 5–6 days of progenitor differentiation in the presence of M-CSF, macrophages represented the most prominent cell populations in YS- as well as BM-derived Hoxb8 cells (Figure 2a,b). Surface expression of immune cell markers increased with differentiation, as indicated by median fluorescence intensities (Appendix A). A small population of floating and loosely adhering cells represented undifferentiated progenitor cells, as well as a minor proportion of neutrophils characterized by LY6G expression. However, medium exchange and simple washing allowed the removal of these cells before macrophage harvest (Figure 2a). Next, we aimed to determine whether Hoxb8 macrophage progenitors differentiate in vivo. We generated CD45.2 YS Hoxb8 cells and transferred them into the peritoneal cavity of CD45.1 mice (male recipients) (Figure 2c–e). Progenitors differentiated into mature macrophages under physiological conditions in vivo within one week, as indicated by surface expression of F4/80. While YS Hoxb8 progenitors almost exclusively generated macrophages, BM Hoxb8 progenitors produced a larger fraction (approximately 25%) of neutrophils (Figure 2d,e), indicating differences in macrophage lineage commitment.

To address cell death properties of Hoxb8 macrophages, we determined cellular responses to ultraviolet (UV) rays and recombinant Fas ligand (SuperFasL). YS and BM Hoxb8 macrophages were similarly susceptible to apoptosis and necrosis induced by UV radiation as well as death receptor ligation by SuperFasL (Figure 2f). These findings support the notion that classical cellular properties, i.e., proliferation, differentiation and death, are comparable between YS and BM macrophages.

### 3.5. Transcriptome Indicates Inflammatory Properties of BM Hoxb8 Macrophages

As phenotypic and morphological characterizations did not show differences between mature YS- and BM-derived Hoxb8 macrophages, we next conducted a transcriptome analysis to compare both cell populations under identical environmental conditions. A large proportion of genes overlapped as expected, and differentiated macrophages of YS and BM origin clustered together in the principal component analysis (Appendix A). Nonetheless, we identified several genes that were differentially expressed between YS and BM macrophages (Figure 3a–e). Specifically, RNA abundance of *Ripk3* and *Pycard* was increased in BM Hoxb8 macrophages pointing towards a predominant role in inflammatory processes with involvement of the inflammasome pathway (Figure 3a,b). Similarly, C–X–C motif chemokine ligand (*Cxcl*) 10 and *Cxcl11*, as well as inducible nitric oxide synthase (*Nos2*), were upregulated in BM Hoxb8 macrophages. Additionally, in BM Hoxb8 macrophages we identified an increased abundance of apoptosis-associated genes, such as *Tgm2*, which promotes leukocyte apoptosis [19]. *Fam129a* encodes for the apoptosis-regulating protein Niban [20,21], which has recently been identified in human atherosclerotic plaques in a macrophage subpopulation and gives strong indications of being of BM origin [22].

In contrast, YS-derived Hoxb8 macrophages expressed higher RNA levels of *clusterin* (apolipoprotein J), which has been associated with apoptotic cell clearance and matrix reorganization. These processes are linked to tissues homoeostasis, a common function of M2-like macrophages [23,24]. Similarly, the transcription factor *Mafb*, which is known to promote anti-inflammatory polarization and cholesterol efflux, was upregulated in YS-derived Hoxb8 macrophages [25]. The triggering receptor expressed on myeloid cells 2 (*Trem2*) has recently been identified as a marker for macrophage subpopulations of potential YS origin. At the protein level, clusterin has been shown to act as a binding partner of TREM2 and is significantly expressed in brain microglia, which are exclusively of YS origin [26,27,28]. In line with this, we identified the upregulation of Sal-like proteins (SALL) 1 and 3, which have been linked to microglia development [29]. Microtubule-associated tumor suppressor candidate 2 (*Mtus2*) is among the most upregulated genes in YS Hoxb8 macrophages (Figure 3c) and has recently been identified as a binary protein interaction partner of M-CSF [30]. We further investigated RNA expression levels after stimulation with either IL4 or LPS under standardized in vitro conditions. Cell-type specific differences between YS and BM macrophages in top regulated genes were in part independent of cytokine stimulation, such as expression of *Ctsz* and *Irak3* (Figure 3d,e; Appendix A). This supports the role of cell-intrinsic macrophage programs.

Differences in the inflammatory profiles between BM and YS Hoxb8 macrophages were corroborated by Gene Set (GSEA) and Gene Ontology (GO) term enrichment analysis (Appendix A). They further suggested differences in cellular respiration, which might be linked to distinct metabolic properties.

### 3.6. BM-Derived Hoxb8 Macrophages Display Higher Glycolytic Activity

Since the gene enrichment analysis indicated differences in oxidative phosphorylation (Appendix A), we interrogated this aspect at the functional level by measuring extracellular flux. While basal respiration of BM and YS macrophages showed no differences, in-depth partitioning of oxygen consumption rates (OCRs) and extracellular acidification rates (ECARs) revealed differences in bioenergetic properties. BM macrophages showed lower rates of protein leak respiration despite higher rates of uncoupler-induced maximum substrate oxidation capacity, resulting in higher mitochondrial efficiency in the resting state (coupling efficiency, CE) and in maximally activated mitochondria (Figure 3f–i). Higher ECARs reported higher glycolytic activity in BM macrophages in steady state as well as under LPS and IL4 treatment (Figure 3j,k). Plotting OCR against ECAR revealed higher overall metabolic activity in BM versus YS-derived Hoxb8 macrophages and showed that BM-derived macrophages more pronouncedly switched from oxidative phosphorylation to glycolysis (‘Warburg-like effect’) under LPS stimulation (Figure 3l). IL4 stimulation did not alter macrophage metabolism under the conditions analyzed (Figure 3k,l; Appendix A). In summary, BM-derived Hoxb8 macrophages presented higher glycolytic activity supporting an inflammatory phenotype.

### 3.7. Proteome Reveals Origin-Specific Differences among Macrophages

We next conducted an in-depth proteome analysis by quantitative mass spectrometry to further compare both YS and BM macrophages under identical environmental conditions. Besides a large proportion of overlapping proteins, as expected, we identified proteins with differential abundance between both cell populations (Figure 4a). BM Hoxb8 macrophages expressed high levels of the inflammasome activator apoptosis-associated speck-like protein containing a CARD (PYCARD, also referred to as ASC) and proteins of the tumor necrosis factor (TNF) signaling pathway such as receptor-interacting serine/threonine-protein kinase 3 (RIPK3). Further in line with the transcriptome analysis, we determined high abundance of the apoptosis-associated enzyme TGM2 [31]. Intercellular adhesion molecule 1 (ICAM1), an important adhesion molecule involved in leukocyte migration, showed increased abundance in BM macrophages. Finally, lipoprotein lipase (LPL), Myc box-dependent-interacting protein 1 (BIN1), oxysterol-binding protein 1 (OSBP), the leucine-rich repeat-containing protein 59 (LRRC59) and protein Niban (FAM129A) displayed high abundance in BM Hoxb8 macrophages (Figure 4a). Thus, this analysis recapitulated the differential RNA expression of YS and BM macrophages on the proteome level and supports differences in macrophage inflammatory properties and their response to inducers of cell death. YS Hoxb8 macrophages showed increased abundance of coagulation factor 5, the lysosomal protein galactosamine (N-Acetyl)-6-sulfatase (GALNS) and ADP-dependent glucokinase (ADPGK), a protein of the glycolysis pathway (Figure 4a). Chitinase-like protein 3 precursor (CHIL3), a protein which has been associated with alternatively activated macrophages, was slightly elevated in BM-derived Hoxb8 macrophages, whereas after lipopolysaccharide (LPS) stimulation CHIL3 was identified among the top ten upregulated proteins in YS-derived macrophages (Figure 4a,b). In addition, Acod1 was upregulated in YS macrophages at both gene and protein levels in response to LPS (Appendix A). Encoding for aconitate decarboxylase 1, it is considered a regulator of immunometabolism in inflammatory conditions [32].

LPS-stimulated BM-derived Hoxb8 macrophages showed an increased abundance of sequestosome-1 (SQSTM1), which has been associated with autophagy. In addition, TGM2 and CD14 expression significantly increased after LPS in BM-derived macrophages. CD14, in association with toll-like receptor 4 (TLR4), is involved in the inflammatory response to LPS stimulation, which is a process commonly associated with M1-like macrophages [33] (Figure 4b,g) profiler enrichment analysis indicated some functional differences in response to LPS stimulation, such as regulation of the spliceosome and of metabolic processes in BM Hoxb8 macrophages (Appendix A). In summary, the above findings, such as the increased abundance of RIPK3 and PYCARD as well as of monocyte differentiation antigen CD14 after stimulation, strengthened the differential regulation of inflammatory properties in BM Hoxb8 macrophages. To obtain further insights into the associated pathways, we carried out multiplex cytokine assays and specifically addressed inflammasome activation.

### 3.8. Inflammasome Activation Is Abrogated in YS-Derived Hoxb8 Macrophages

First, we studied the response of Hoxb8 macrophages to LPS and IL4 in a multiplex cytokine analysis. Interestingly, cytokine release was similar between YS and BM macrophages (Figure 4c). The increased abundance of PYCARD, an important adaptor protein associated with inflammasome function, at both the RNA and protein level, drove us to further investigate this pathway in YS and BM macrophages. Cytosolic pattern recognition receptors activate the inflammasome through the canonical pathway leading to caspase-1 activation with subsequent release of IL1β, IL18 and Gasdermin D. To investigate canonical activation of the NLRP3 inflammasome pathway, we primed Hoxb8 macrophages with LPS and stimulated them with crystals [34]. Both cholesterol (CH) and monosodium urate (MSU) crystals induced the release of IL1β from LPS-primed BM macrophages (Figure 4d–g). As expected, IL1β secretion was absent in caspase 1/11-deficient BM macrophages, indicating a dependence on caspase activation in this pathway (Figure 4e). Notably, IL1β secretion from LPS-primed crystal-stimulated YS Hoxb8 macrophages was low, indicating a low level of inflammasome activation (Figure 4d–g). We confirmed our findings by protein expression analyses and determined a pronounced inflammasome response upon MSU crystal stimulation in BM Hoxb8 macrophages. In contrast, relative abundance of PYCARD and CASP1 protein was low in YS macrophages (Figure 4f,g; Appendix A). Up to this point, we carried out deep phenotyping and functional assessment in YS and BM macrophages derived from Hoxb8-SCF progenitors. We next determined whether macrophages differentiated from Hoxb8-FL progenitors provided similar inflammasome responses. We generated three independent clones each from mouse BM and E9.5 YS, and differentiated them into mature macrophages in the presence of M-CSF, as performed for Hoxb8-SCF cells. LPS-primed BM macrophages provided a strong release of IL1β following MSU stimulation whereas the inflammasome response of YS Hoxb8-FL macrophages was again minimal (Appendix A). These findings support the notion that the differences in intrinsic cellular programs between YS and BM macrophages are robust and remain detectable across different versions of the Hoxb8 protocol to conditionally immortalize hematopoietic progenitors.

Crystals are classical activators of the canonical inflammasome pathway. To address non-canonical inflammasome activation, we harnessed outer membrane vesicles (OMVs). OMVs are typically produced by Gram-negative bacteria and mediate cytosolic LPS localization leading to robust caspase-11 activation [35]. Stimulation with *E. coli* derived OMVs elicited a pronounced IL1β response in BM macrophages which was abrogated in YS-derived counterparts (Figure 4h). Together, BM-derived but not YS-derived Hoxb8 macrophages exhibited robust NLRP3 inflammasome activation via canonical and non-canonical pathways, highlighting a fundamental difference between the respective macrophage populations.

## 4. Discussion

In this study, we aimed to differentiate the cell-intrinsic programs and functions of YS and BM macrophages under defined conditions in vitro independently of the organ environment. Scientists have been hampered in addressing this question due to adaptation and re-programming of macrophages within tissues. To overcome these limitations, we established YS and BM Hoxb8 progenitor cells, which can be cultured and differentiated to mature macrophages as needed.

Through a broad variety of experimental approaches, including immunofluorescence analysis, flow cytometry, proteomics, transcriptomics, chemokine analyses and functional assays, we were able to draw the following conclusions: Macrophage progenitors expand and differentiate within 5 days to mature macrophages independently of their cellular origin. These macrophages express a very similar panel of basic macrophage markers and they are equally capable of phagocytosis. However, upon a deeper look into the transcriptomes and proteomes of YS and BM macrophages, cell-intrinsic differences became apparent. BM-derived Hoxb8 macrophages expressed higher levels of proteins associated with inflammatory processes, including CD14 (LPS co-receptor), PYCARD (inflammasome), RIPK3 (necrosome), and SQSTM1 (autophagy). These findings characterize BM-derived macrophages as pro-inflammatory immune cells, with some features being reminiscent of M1-like macrophages. YS-derived macrophages in comparison are less inflammatory. In the molecular profiling we identified CHIL3 as a signature protein in these cells, which has been associated with alternative macrophage activation in vivo [36,37,38]. The differences in inflammatory functions of BM and YS macrophages are accompanied by altered metabolic properties. As shown by extracellular flux measurements, BM cells exhibited higher glycolytic levels in steady state and upon LPS stimulation. By contrast, YS-derived macrophages utilized more oxidative phosphorylation to supply their energy demands, a metabolic process linked to anti-inflammatory and homeostatic functions [39,40]. Our findings are in line with and extend a recent study on embryonic macrophages, in which reduced expression of NLRP3 inflammasome-associated genes have been described in YS macrophages when compared to fetal liver (FL) macrophages, the latter displaying a proinflammatory gene signature as well as increased levels of Sqstm1 [41].

We show here that protein abundance of PYCARD and Caspase-1, which are essential components of the inflammasome machinery, was low in YS macrophages. Consequently, IL-1β secretion upon canonical (crystals) and non-canonical (OMV) inflammasome activation was robust in BM macrophages but virtually absent in YS macrophages. Notably, PYCARD is constitutively expressed and not induced by inflammatory signaling [42], underlining the importance of cell-intrinsic macrophage programs determining the expression of inflammasome components. Together, YS macrophages are less well equipped with inflammatory machinery, which abrogates inflammasome-associated responses.

The non-inflammatory signature of YS macrophages could have various biological implications. For instance, organ growth and maturation of the developing embryo requires the support of macrophages that promote angiogenesis or provide phagocytic functions rather than proinflammatory properties [43,44]. In addition, limited inflammasome activation in YS macrophages might play a role in the development of immunologic tolerance towards food antigens or against normal bacterial flora [45]. In adulthood, YS macrophages continue to provide functions that support tissue homeostasis and regeneration [46], indicating that cell-intrinsic programs can be maintained in life. In the future, it will be of interest to determine conditions that induce functional reprogramming of YS-derived macrophages as well as its impact on organ functions or priming towards the development of inflammatory conditions. In line with previous work using Hoxb8-mediated conditional immortalization, these macrophages will enable further studies into their biological functions as well as their genetic modifications [47,48].

## 5. Conclusions

We established a Hoxb8 model system to identify unique properties of macrophages based on their cellular origin under defined conditions in vitro. This setting allowed us to define cell-intrinsic programs in the absence of organ microenvironments, which govern and modulate macrophage phenotype and functions. In general, macrophages are characterized by high cellular plasticity regulated by their surrounding milieu. However, ontogeny-associated properties persist and could potentially be targeted by therapeutic approaches.

## Figures and Tables

**Figure 1 cells-10-03564-f001:**
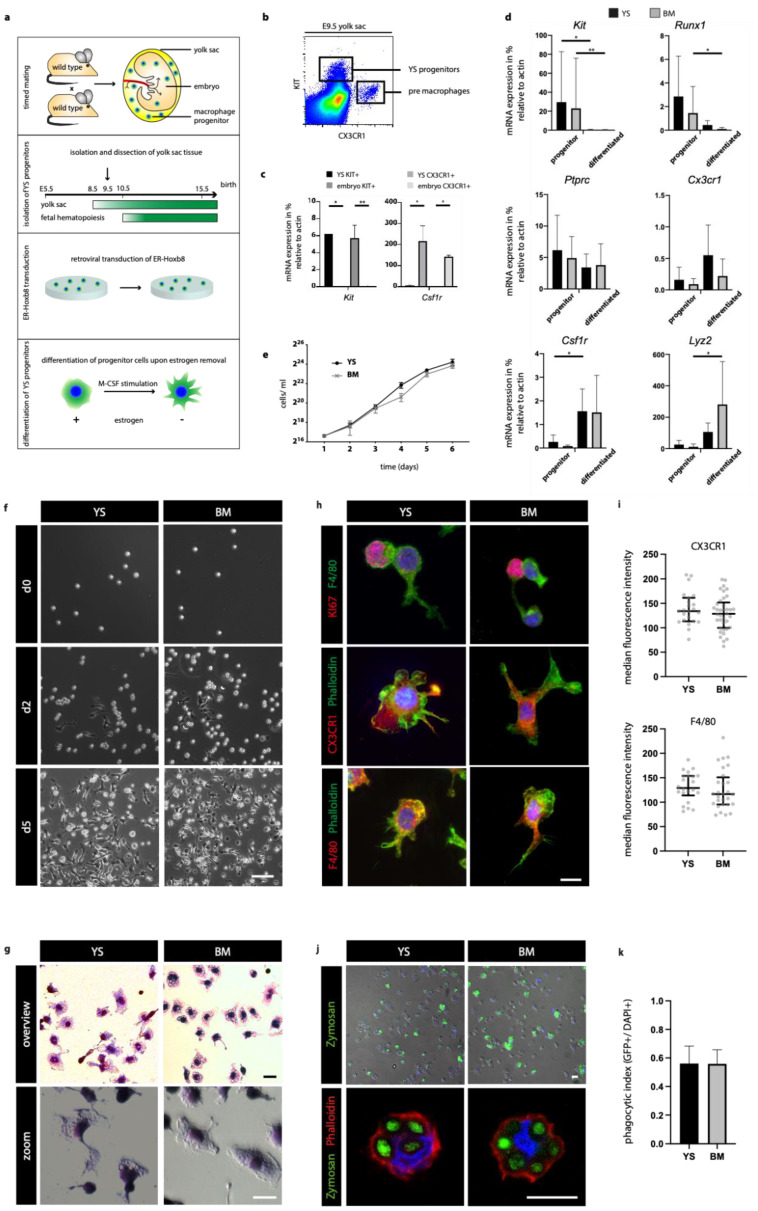
Generation of functional Hoxb8 YS macrophages. (**a**) Schematic graph illustrating the process of Hoxb8 cell line generation. (**b**) Flow cytometry of E9.5 hematopoietic progenitors in the yolk sac (YS) (representative plot, *n* = 4). (**c**) mRNA expression analysis of YS and embryonic KIT+ progenitors and CX3CR1+ pre-macrophages (*n* = 3). (**d**) mRNA expression analysis of Hoxb8 progenitors and differentiated Hoxb8 macrophages for indicated genes in percent (%) relative to beta actin expression (*n* = 3). (**e**) Growth curve for YS and BM Hoxb8 progenitors (*n* = 3). (**f**) Bright field images of Hoxb8 cell lines during the process of macrophage differentiation. (**g**) Microscopic images of Hoxb8 macrophages stained with May–Grünwald–Giesma (*n* = 3). (**h**) Immunofluorescence analysis with indicated antibodies (nuclei in blue, Hoechst) and (**i**) quantification of median fluorescence intensity of the markers Cx3cr1 and F4/80. Median +/− IQR. (**j**) Phagocytosis assay with pHrodo Zymosan bioparticles (GFP, green) after 1 h of incubation. (**k**) The ratio of phagocytic (GFP+) to DAPI+ macrophages is indicated (3 independent experiments). Panels show representative images. Scale bars represent 100 µm (**f**), 20 µm (**g**,**h**,**j**). (**) *p* < 0.01, (*) *p* < 0.05.

**Figure 2 cells-10-03564-f002:**
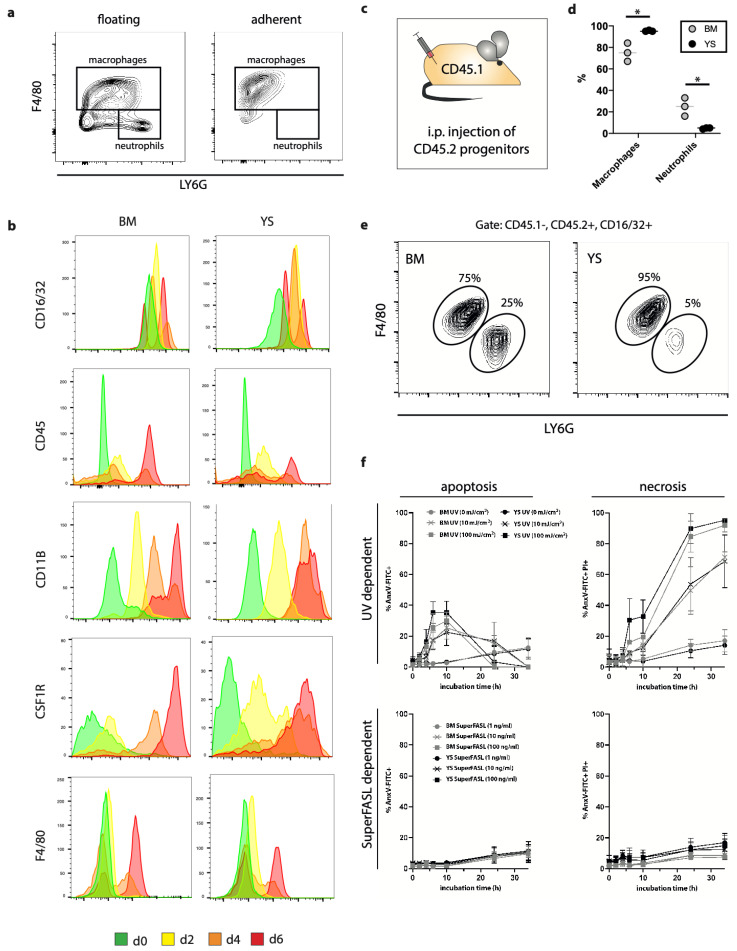
Expression profile and survival of *Hoxb8* macrophages. (**a**) Flow cytometry analysis of floating or adherent differentiated (day 5) Hoxb8 BM cells labeled with indicated antibodies (representative plot, *n* = 3). (**b**) Flow cytometry analyses of Hoxb8 progenitors in the process of differentiation towards macrophages (d0 to d6) labeled with indicated antibodies (representative experiment of *n* = 3). (**c**) Schematic graph for intraperitoneal Hoxb8 progenitor application. (**d**–**f**) Flow cytometry quantification of in vivo differentiated Hoxb8 macrophage and neutrophil populations obtained by peritoneal lavage after one week. Live CD45.1− CD45.2+ CD16/32+ single cells were gated, then macrophages and neutrophils were separated by F4/80 and LY6G intensities. Results are displayed as (**d**) relative quantities of differentiated cell populations; (**e**) corresponding contour plots (representative experiment of *n* = 3). (**f**) Quantification of UV and SuperFASL dependent apoptosis and necrosis in differentiated Hoxb8 macrophages. *n* = 3. Unpaired, two-tailed *t*-test. (*) *p* < 0.05.

**Figure 3 cells-10-03564-f003:**
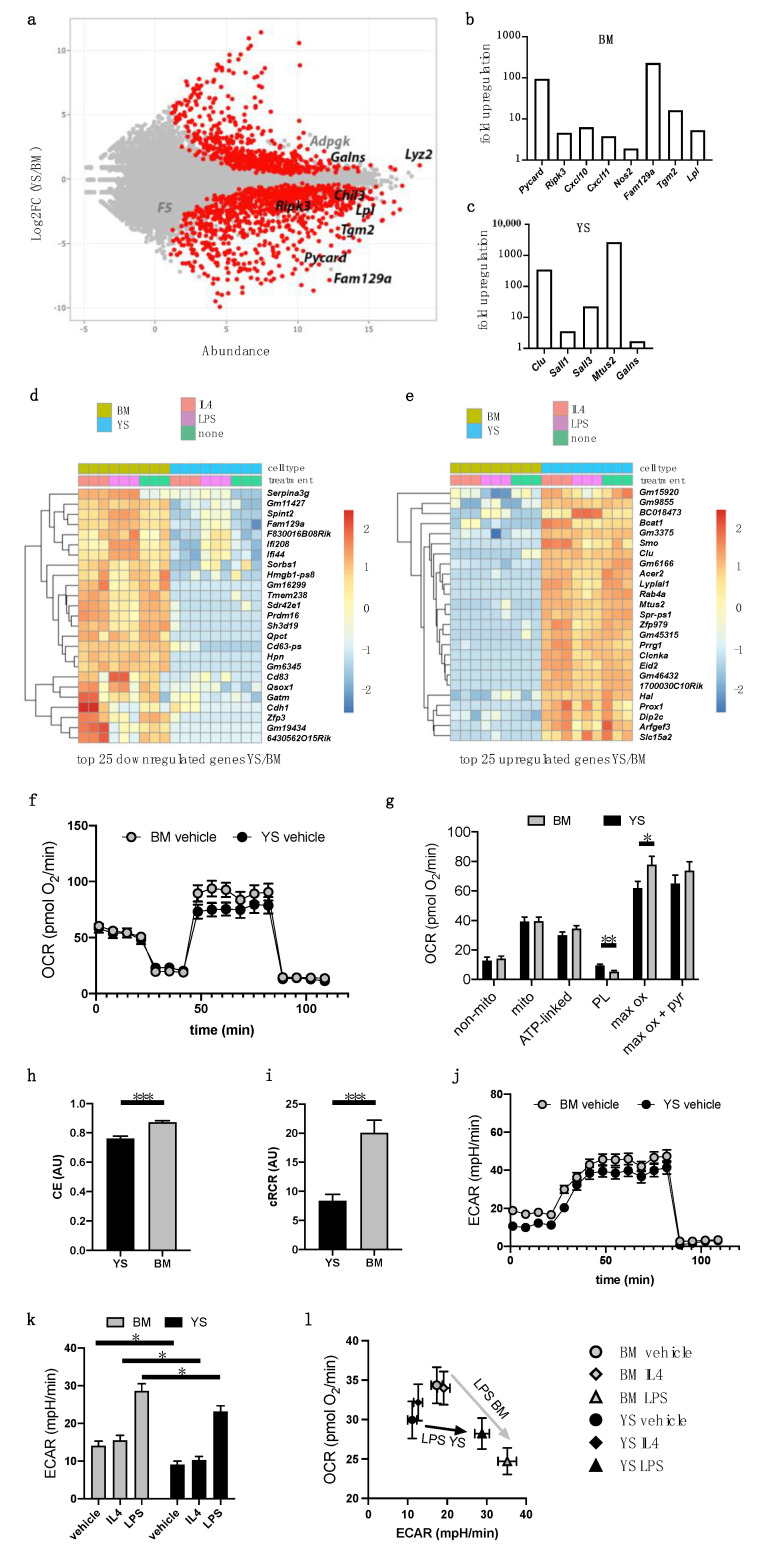
Transcriptome and extracellular flux analysis. (**a**–**e**) RNA expression analysis of differentiated YS and BM Hoxb8 macrophages. (**a**) MA plot of macrophage gene expression indicating fold changes (log2) of YS versus BM plotted against abundance (logTPM). Genes are marked in grey (not significant) or black (significant). (**b**,**c**) Fold upregulation of selected genes in (**b**) BM- and (**c**) YS-derived macrophages. Top 25 (d) upregulated and (**e**) downregulated genes of YS versus BM Hoxb8 cells without treatment or stimulated with IL4 or LPS. (**f**–**l**) Bioenergetic profiling of Hoxb8 macrophages either stimulated or not stimulated with IL4 or LPS for 4 h using XF96 extracellular flux analyzer as described in the Methods section. All data represent the mean of 26–29 wells measured on three independent experimental days and are normalized to 130 ng ds DNA/well, two-way ANOVA, followed by Sidak post hoc test. (**f**) Oxygen consumption rates (OCR) traces of unstimulated BM and YS macrophages using respiratory inhibitors to probe bioenergetic modules. (**g**) Respiratory modules of mitochondrial energy transduction in unstimulated and stimulated cells. (**h**) Mitochondrial efficiency. Coupling efficiency (CE) is the respiratory fraction driving ATP synthesis at resting state; (**i**) cellular respiratory control ratio (cRCR) is determined using proton leak and uncoupler-induced respiration. (**j**) Extracellular acidification rate (ECAR) traces of unstimulated BM and YS macrophages using specific inhibitors. (**k**) ECAR linked to glycolytic activity of unstimulated and stimulated macrophages. (**l**) Glycolytic ECAR plotted against ATP-linked OCR to depict global changes in cellular energy metabolism, revealing the metabolic switch induced by LPS in BM Hoxb8 macrophages. (***) *p* < 0.001, (**) *p* < 0.01, (*) *p* < 0.05.

**Figure 4 cells-10-03564-f004:**
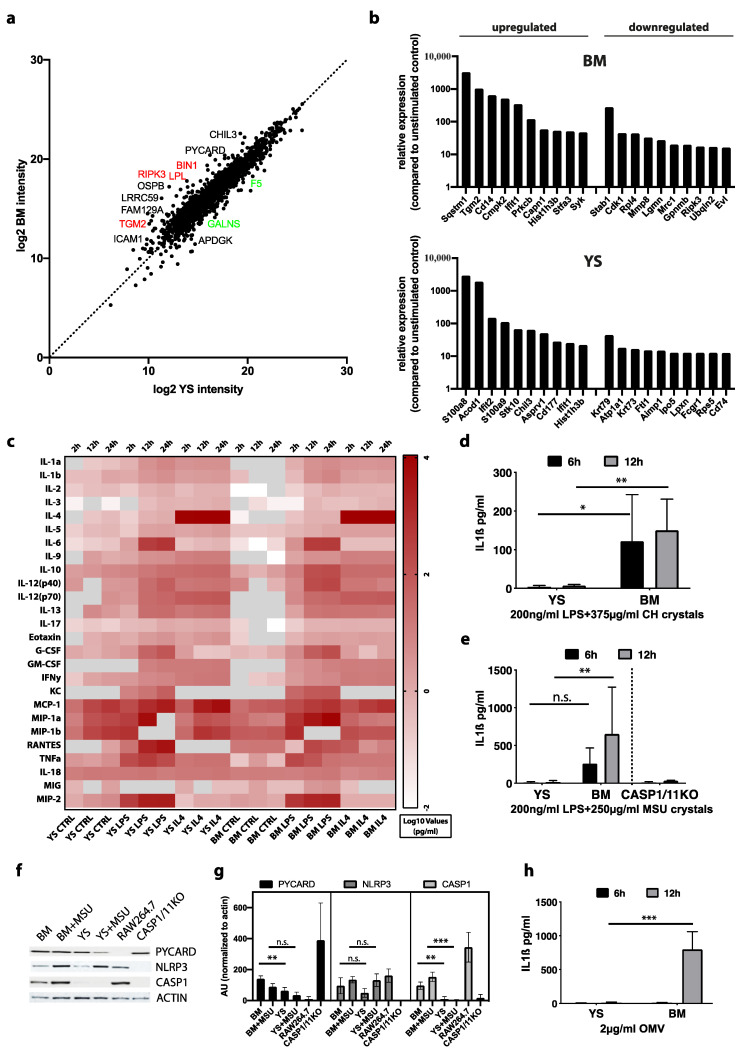
Inflammasome activation in Hoxb8 macrophages. (**a**,**b**) Proteome analysis of day 5 differentiated Hoxb8 macrophages. (**a**) Protein intensities (log2) of BM versus YS macrophages are indicated. Selected, differentially abundant proteins were annotated and colored in red (BM) or green (YS). (**b**) Top 10 up- and downregulated proteins after LPS stimulation. (**c**) Cytokine expression analysis by Multiplex ELISA in untreated (CTRL) and LPS or IL4 stimulated samples. *n* = 3. Pearson’s correlation. (**d**–**e**) IL1β expression analysis by ELISA in LPS-primed crystal stimulated (6 h or 12 h) YS and BM macrophages. One-way ANOVA with Bonferroni post hoc test. (**d**) Stimulation with cholesterol (CH) crystals. *n* = 8. (**e**) Stimulation with monosodium urate (MSU) crystals. *n* = 7. BM Hoxb8 macrophages from Caspase 1/11-deficient mice served as control. *n* = 6. (**f**,**g**) Detection of PYCARD, NLRP3 and CASP1 protein expression in YS and BM Hoxb8 macrophages treated with 250 ug/mL MSU crystals for 24 h (**g**) with corresponding quantifications (*n* = 3). Beta actin served as loading reference. RAW264.7 cells served as PYCARD-deficient control. One-way ANOVA with Bonferroni post hoc test. (**h**) IL1β expression analysis by ELISA of macrophages stimulated with *E. coli* outer membrane vesicles (OMV) (*n* = 3). One-way ANOVA with Bonferroni post hoc test. (***) *p* < 0.001, (**) *p* < 0.01, (*) *p* < 0.05.

## Data Availability

The authors declare that all data supporting the findings of this study are available within the paper and its Appendix A files. The RNA-seq raw data have been deposited in NCBI GEO under the accession code GSE176409. The mass spectrometry proteomics data have been deposited to the ProteomeXchange Consortium via the PRIDE (Perez-Riverol, Csordas et al., 2019) partner repository with the dataset identifier PXD026922. There are no restrictions regarding data availability.

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
