# Peer review of "Differences in Cell-Intrinsic Inflammatory Programs of Yolk Sac and Bone Marrow Macrophages"

_cells, 2021, doi:10.3390/cells10123564_

Round 1
Reviewer 1 Report
In the manuscript by Elhag et.al., the authors investigate the differences between the yolk sac and bone marrow macrophages. While no major differences are observed in properties of macrophages under resting state, the YS and BM macrophages display differences in metabolic regulation, inflammatory marker expression, and inflammasome activation upon LPS stimulation. The study is well designed and investigates multiple aspects of macrophage properties using a variety of high-throughput techniques. The research is of interest, timely, and addresses an important aspect of macrophage origin and development. The study could be more relevant if the suggested analysis is performed to understand the comprehensive differences between different polarized states. The manuscript is well suited for publication. However, I have the following concerns that should be addressed before publication:
- In Fig1H, the mean fluorescence intensity of different markers should be calculated and reported in the data.
- In Fig 1I, the phagocytic index should be calculated and shown.
- The FACS data shown in Fig 2B should be quantified.
- The statistical analysis is missing for figure 2D.
- In line 427, the authors state that the differentiation is comparable between YS and BM macrophages, however, the data shown in Fig 2C-E shows difference in YS and BM macrophages in the in vivo differentiation experiment. The data is not supporting the conclusions drawn from the data and this discrepancy should be resolved.
- The authors have identified top upregulated and downregulated genes between YS and BM macrophages in the transcriptomics data. In this analysis, the differences between the different polarized states are not highlighted. The authors should also perform individual comparisons of IL4 and LPS stimulated YS and BM macrophages to find differentially regulated genes between polarized states.
- The representative ECAR plots of IL4 and LPS stimulated macrophages should also be included in the manuscript along with the data shown in Fig 3K and L.
- While the authors have discussed some common protein/ genes commonly regulated in both the transcriptomic and proteomic data for LPS stimulated macrophages. However, the authors should perform a comprehensive analysis to identify commonly up- or downregulated proteins between both datasets for LPS stimulated YS and BM macrophages individually.
- The statistical analysis in Fig 4D, E, G, and H is calculated by Student’s two-tailed t-test which is not appropriate as there are multiple groups in the dataset. The authors should perform statistical analysis by One-way ANOVA test.
- The terms in vitro and in vivo are not italicized in the manuscript and should be corrected.
Reviewer 2 Report
Elhag and colleagues evaluated genetic differences of hematopoietic progenitors from yolk sac and bone marrow-derived macrophages using HoxB progenitors and maturing them into macrophages in order to further describe cell regulation, phenotypic differences and programing between the two cell subsets. They noted similarities in cell growth, while noting differences in cell metabolism and inflammatory markers .
I would like to commend the authors in their thorough description of their methodology as well as providing the necessary supplemental material showing their work and that allows reproducibility of their work.
This is interesting work that shows genetic differences between two linages. One thing I wanted to ask the authors is if they considered future studies evaluate epigenetic regulations between YS and BM macrophages (PU.1-bound chromatin) for induction of pro-inflammatory mediators.
Minor comments, perhaps the authors could enhance the manuscript by providing additional background in the discussion section for previous Hox8B-conditional-immortalization of macrophage progenitors ie. https://onlinelibrary.wiley.com/doi/10.1002/eji.201040962
Out of curiosity, Did the authors evaluated nitric oxide level in YS-macrophages? or attempt to elicit anti-inflammatory responses by BM and YS Macs using IL-4?
This manuscript will allow for further characterization of two critical subsets of macrophages.
